# Diurnal Variation in Transport and Use of Intracellular Leaf Water and Related Photosynthesis in Three Karst Plants

Xiaojie Qin [1], Deke Xing [1,*], Yanyou Wu [2], Weixu Wang [1], Meiqing Li [1] and Kashif Solangi [1]

1    School of Agricultural Engineering, Jiangsu University, Zhenjiang 212013, China
2    State Key Laboratory of Environmental Geochemistry, Institute of Geochemistry, Chinese Academy of Sciences, Guiyang 550081, China
*    Correspondence: xingdeke@ujs.edu.cn; Tel.: +86-511-88797338

**Abstract:** Except for transpired water, the intracellular water stored in leaves accounts for only 1–3% of the water absorbed by roots. Understanding water transport and use, as well as the related photosynthetic response, helps with determining plant water status and improving the revegetation efficiency in fragile karst habitats. In this study, we conducted experiments on 8 year old naturally growing plants of *Coriaria nepalensis* Wall., *Broussonetia papyrifera* (L.) Vent., and *Elaeocarpus decipiens* Hemsl. in karst areas. We determined the diurnal variations in leaf electrophysiology, water potential, gas exchange, and chlorophyll fluorescence parameters. The results indicated that *C. nepalensis* plants maintained a high photosynthetic rate, with a high root water uptake ability and leaf intracellular water-holding capacity (LIWHC). The stomata quickly closed to conserve water within cells and protect the photosynthetic structure. *B. papyrifera* maintained stable intracellular water transport rate (LIWTR), and the photosynthetic efficiency was increased with increasing intracellular water-use efficiency (LIWUE). *B. papyrifera* also maintained its photosynthesis by efficiently using the transpired water when the LIWHC was increased. The inter- and intracellular water in the leaves of *E. decipiens* remained stable, which could be attributed to the leathery leaves and its high water-holding capacity. The photosynthesis of *E. decipiens* was low and stable. Compared with the high photosynthesis, high transpiration, and low instantaneous water-use efficiency (WUE$_i$) pattern in *C. nepalensis* plants, *E. decipiens* plants exhibited low photosynthesis, low transpiration, and low WUE$_i$, whereas *B. papyrifera* plants presented high photosynthesis, low transpiration, and high WUE$_i$. Plants in karst regions change their transport and use of intracellular leaf water to regulate the photosynthetic performance, which differs among different plant species.

**Keywords:** electrophysiology; stomatal conductance; water-use efficiency; chlorophyll fluorescence; adaptability

## 1. Introduction

The karst landscape in Guizhou province, China, is one of the largest continuous landscapes in southwestern China [1]. This karst landscape is located in a subtropical monsoon climate zone and has an annual average temperature of 14 °C and precipitation of 1300 mm [2]. In recent years, rocky karst desertification is becoming increasingly serious due to frequent human activities, resulting in a shallow and discontinuous soil layer, which has gradually decreased the vegetation cover and soil water retention capacity [3]. The vegetation in these regions is mainly composed of uneven grasses and shrubs [4,5], which has produced an uneven surface hydrologic permeability distribution. Despite receiving enough precipitation, the plants in these regions usually suffer from temporary water stress. Most importantly, the droughts within karst regions are highly heterogeneous [6–8]. Therefore, plant adaptability to the heterogeneous drought environments must be studied, allowing the selected plant species to be matched with fragile karst habitats and the vegetation restoration efficiency to be increased.

Water is crucial for plant growth and development, but water-use strategies differ among plant species [9,10]. Stomata are the first line of defense against drought stress [11–13] and play an important role in the water transport within plants by regulating the apertures [14]. With opened stomata apertures, a high transpiration rate (E) is accompanied by strong photosynthetic $CO_2$ assimilation. However, a high E decreases the leaf water potential ($\Psi_L$) and can damage the mesophyll, thus depressing photosynthesis [12]. Stomatal closure can reduce water loss and prevent irreversible damage caused by water deficits to the photosynthetic structures [15]. Photosynthetic performance under water stress can be used for investigating the water-use traits in plants [16]. Photosynthesis is a dynamic reaction process. The diurnal variations in photosynthetic parameters can indicate the photosynthetic capacity of a plant [17]. Water stress also influences the chlorophyll fluorescence parameters, which are closely related to each reaction process in photosynthesis [18]. Chlorophyll fluorescence parameters can reflect the characteristics of plant internality [19]. Photochemical quenching (qP) represents the openness of the photosystem II (PSII) reaction center [20,21], whereas nonphotochemical quenching (NPQ) is a plant self-protective mechanism against excess light energy. Under water stress conditions, the daily average minimal fluorescence ($F_o$) increases, while the maximum fluorescence yield ($F_m$) and primary light energy conversion efficiency ($F_v/F_m$) of PSII decrease [22]. Therefore, the chlorophyll fluorescence can be used to nondestructively determine the internal reaction process of the photosynthetic structure and evaluate the response of a plant to adversity [20].

Most (~97%) of the water absorbed by roots is dissipated through transpiration, while only a small part (1~3%) is stored within leaf cells, which directly determines the photosynthesis, growth, and other metabolic processes of the plant [14]. Photosynthesis is directly correlated with intracellular water transport and use. When plants are subjected to water deficits, changes in leaf intracellular water alter the concentrations of cell sap and electrolytes (i.e., ions, ion groups, and electric dipoles in cells) [23]. The cell membrane, having strictly selective permeability, also influences the concentration of intracellular electrolytes, which mainly exist in the vacuoles and cytoplasm [24]. The cytoplasm contains numerous organelles with specific membranes, and electrical features vary across the organelles, vacuoles, and cytoplasm, which occupy most of intracellular space and can be regarded as resistors, whereas the plasma membrane shows capacitance. The electrolyte solutions on the two sides of a cell membrane form a specific conductive state. A mesophyll cell can be modeled as a concentric sphere capacitor due to the abovementioned special composition and structure [24,25]. The water metabolism in leaves alters the electrolyte concentration and changes the corresponding electrophysiological parameters. Therefore, electrophysiology is increasingly being used to diagnose the status of plant water [26,27]. Electrophysiological parameters, such as physiological capacitance and impedance [23,24,27], are related to the change in solute concentration within leaf cells; they can reflect the transport of intracellular dielectric substances and be used to detect the dynamic characteristics of intracellular water. The findings of studies on the transport and use of intracellular water stored in the leaves, rather than the transpired water, can help to accurately determine the water status of plants.

*Coriaria nepalensis* Wall. is a deciduous shrub with strong reproductive capacity and papery or thin leathery leaves [28,29]. It is widely distributed in the Shanxi, Sichuan, and Gansu provinces in China. *C. nepalensis* has a strong adaptability and can resist drought and barren environments. This type of plant grows well in neutral alkali soil. *Broussonetia papyrifera* (L.) Vent. is a deciduous tree with a shallow root system, wide lateral root distribution, and papery leaves [30]. *B. papyrifera* also has a strong adaptability and can resist to drought and barren environments, thus being widely distributed. *Elaeocarpus decipiens* Hemsl. is an evergreen tree that is slightly shade-tolerant, with a developed root system, strong germination, and leathery leaves [31]; it has high ornamental value and is suitable for planting in the mining areas due to its strong tolerance to $SO_2$ [32]. These three plant species all grow well in karst areas, but the water transport and use traits of

these plants remain unknown. As such, in this study, we selected *C. nepalensis*, *B. papyrifera*, and *E. decipiens* as the experimental object, and we measured the diurnal variations in electrophysiology, $\Psi_L$, gas exchange, and chlorophyll fluorescence parameters of these plants. We calculated and analyzed the dynamic characteristics of leaf intracellular water transport as a function of the electrophysiological parameters. Our aims in this study were to (i) analyze the relationship between leaf intracellular water and photosynthesis, (ii) explore the diurnal variation in transport and use of intracellular leaf water in different plants, and (iii) compare the adaptabilities of the three plant species to a karst environment. The results provide a basis for selecting appropriate plant species matched to the heterogeneous karst environment during the vegetation restoration in fragile karst habitats.

## 2. Materials and Methods

### 2.1. Study Area

We conducted this study at the National Observation Station of Karst Ecosystem (Puding Station) in Puding County, Guizhou province, southwest China (26°22′07″ N, 105°45′06″ E). This experimental site was 1158 m above sea level with a humid subtropical monsoon climate, and an average annual temperature of 15.1 °C, with a maximum of 34.4 °C in summer and a minimum of –11.0 °C in winter. The average annual precipitation was 1397 mm·year$^{-1}$. The vegetation types were mainly degraded rattan shrub and secondary evergreen and deciduous broadleaved forest. The soil type was mainly lime soil with a loam texture. Hang et al. (2018) reported that the soils in this site were characterized by high calcium, phosphorus deficiency, high bicarbonate, and alkaline properties [33]. We conducted the experiment in July 2021, which was a period with a relatively high temperature and water deficiency.

According to the planning and construction of the National Observation Station and the branch growth of the selected plants, we selected the leaves of approximately 8 year old *C.nepalensis*, *B. papyrifera*, and *E. decipiens* plants that were naturally growing around Puding station as the experimental subject. We used five leaves from five different randomly selected plants for each parameter determination in each plant species. The abovementioned plant species differed in distribution, adaptive habitats, and biological traits (Table 1). We recorded the measurements on the fourth or fifth fully expanded leaves that were growing well and uniformly. We measured the parameters every 2 h from 8:00 a.m. to 6:00 p.m. to study their diurnal variations.

**Table 1.** Comparison between *C.nepalensis, B. papyrifera, and E. decipiens.*

| Plant Species | Distribution | Adaptive Habitats | Biological Traits |
|---|---|---|---|
| *C. nepalensis* | Yunnan, Guizhou, Sichuan, Hubei, Shanxi, Gansu, Xizang provinces in China; India; Nepal | Drought, low nutrition, neutral alkaline, heliophilous, low temperature | Shrub, fast-growing, leathery leaf, shallow root system with many horizontal and oblique roots |
| *B. papyrifera* | Yellow, Yangtze and Pearl River Basins in China; Vietnam; Japan; India; Malaysia; Thailand; Burma | Drought, low nutrition, waterlog, heliophilous, acidic and neutral soil, chimney, air pollution, limestone | Tree, fast-growing, papery tomentose leaf, shallow root system with wide lateral root distribution |
| *E. decipiens* | Guangxi, Guangdong, Guizhou, Jiangxi, Fujian, Zhejiang, Yunnan, Hunan, Taiwan provinces in China; Vietnam; Japan | Slightly shade-tolerant, liked warmth and humid, acid soil, strong tolerance to $SO_2$ | Tree, fast-growing, leathery leaf, developed deep root system |

### 2.2. Determination of Electrophysiological Parameters and Leaf Water Potential

Electrophysiological parameters are easily affected by environmental stimuli. Therefore, we removed the plant leaves from the branches and immediately immersed them in water for 30 min. As such, we ensured that the cells of the leaves reached a standard and uniform state, which helped to minimize the influence of environmental stimuli on the values of the electrophysiological parameters. Then, we dried the leaf surfaces with tissues, and we clamped the leaves with a custom-made parallel-plate capacitor [24]. We only used the rehydrated leaves for determining the electrophysiological parameters. We recorded the variations in the electrophysiological parameters (i.e., physiological capacitance, resistance, and impedance) with increasing gripping force using an LCR tester (*Model 3532-50, Hioki*, Nagano, Japan). We set the gripping forces to 2.1, 4.1, 6.1, and 8.1 N. The leaves were not to be damaged by these gripping forces, as they were not strong enough. We first placed the leaf between two parallel electrode plates with a diameter of 7 cm, and we applied different gripping forces by adding the same quality iron blocks. We selected three sites on each leaf for recording the electrophysiological parameters at each gripping force, and we calculated the average value of each parameter. We recorded the measurement on five leaves from five different randomly selected plants for each plant species.

We established the coupling models of gripping force and electrophysiological parameters according to the Nernst equation and the law of energy conservation, respectively. We then calculated the leaf intracellular water transport rate (LIWTR) [24], water-holding capacity, and water-use efficiency [23]. The specific calculation formulas (Supplementary File 1) were as follows:

$$\text{LIWTR} = \text{bke}^{-\text{bF}}, \tag{1}$$

$$\text{LIWHC} = \sqrt{(\text{IC})^3}, \tag{2}$$

$$\text{LIWUE} = \frac{\text{d}}{\text{LIWHC}}, \tag{3}$$

where b and k are parameters of the physiological impedance fitting equation, IC (pF) is the leaf physiological capacitance, and d is the specific effective thickness of the leaf.

We detached the leaves from the branches of the naturally growing plants, which were immediately drilled with a hole punch. We then placed the small, drilled disc into a C-52-SF sample chamber; after 6 min balancing, we recorded the $\Psi_L$ data with a dew point microvoltmeter (*Psypro, Wescor*, United States) [34,35]. We measured the $\Psi_L$ at the same position as the abovementioned electrophysiological parameters. We conducted the measurement on five leaves from five different randomly selected plants for each plant species.

### 2.3. Determination of Photosynthetic Parameters

We measured the diurnal variations in photosynthetic parameters with a portable LI-6400XT photosynthesis measurement system (*LI-COR Inc., Lincoln*, NE, United States). We clamped the leaf with a transparent leaf chamber, which was vertically irradiated with natural light. The photosynthetic photon flux density (PPFD) varied from 36.66 to 1989.99 $\mu\text{mol}\cdot\text{m}^{-2}\cdot\text{s}^{-1}$ throughout the day, the air temperature was 33.42 $\pm$ 6.46 °C, and the $CO_2$ concentration was 415.49 $\pm$ 15.26 $\mu\text{mol } CO_2\cdot\text{mol}^{-1}$. We repeated the in situ and nondestructive measurements of photosynthetic parameters five times by randomly selecting five different plants for each plant species. We recorded the net photosynthetic rate ($P_N$, $\mu\text{mol}\cdot\text{m}^{-2}\cdot\text{s}^{-1}$), stomatal conductance (gs, $\text{mmol}\cdot\text{m}^{-2}\cdot\text{s}^{-1}$), and transpiration rate (E, $\text{mmol}\cdot\text{m}^{-2}\cdot\text{s}^{-1}$). We calculated the instantaneous water-use efficiency (WUE$_i$, $\mu\text{mol}\cdot\text{mmol}^{-1}$) as follows [36]:

$$\text{WUE}_i = P_N/\text{E}, \tag{4}$$

### 2.4. Determination of Chlorophyll Fluorescence Parameters

We measured the chlorophyll fluorescence parameters using an IMAGING-PAM modulated chlorophyll fluorescence analyzer (*Heinz Walz GmbH, Effeltrich*, Germany). The PSII reaction center was completely opened after 20 min of dark adaptation before measurement. We irradiated the fully dark-adapted leaves with modulation measurement light (about 0.10 µmol·m$^{-2}$·s$^{-1}$), and we recorded the $F_o$ value. Then, we irradiated the leaves with a saturated light flash (usually about 3000 µmol·m$^{-2}$·s$^{-1}$ or higher for less than 1 s). The primary electron acceptor QA was completely restored in a short time, the PSII completely "switched off", and we measured the Fm. We measured the qP and NPQ of the front of plant leaves with an activation light of 800 µmol·m$^{-2}$·s$^{-1}$. We calculated $F_v/F_m$ as $(F_m - F_o)/F_m$. We conducted the measurement on five leaves from five different randomly selected plants for each plant species.

### 2.5. Statistical Analysis

We analyzed all collected data using SPSS software (*version 22.0, SPSS Inc.*, New York, NY, USA) and SigmaPlot software (*version 10.0, Systat Software Inc.*, California, CA, USA). We fit the coupling relationships between gripping force and the electrophysiological parameters using SigmaPlot software. We compared the parameters at different times with Duncan's multiple comparison at the 5% significance level ($p \leq 0.05$) using SPSS software. The data are reported as the means $\pm$ standard errors ($n$ = 5). We prepared the graphs using Origin 2019 (*Northampton*, MA, USA).

## 3. Results

### 3.1. Diurnal Variations in Leaf Water Potential and Electrophysiological Parameters

The $\Psi_L$ values of *C. nepalensis* between 8:00 and 10:00 a.m. were not significantly different, but were clearly lower than at 2:00 and 6:00 p.m. (Figure 1). The $\Psi_L$ values of *B. papyrifera* at 8:00 a.m. and 12:00 p.m. were significantly higher than those at 10:00 a.m. and 2:00 p.m., and those at 6:00 p.m. were not notably different from the values at 8:00 a.m. and 12:00 p.m. The $\Psi_L$ value of *E. decipiens* at 10:00 a.m. was clearly higher than that between 12:00 and 6:00 p.m., and the values exhibited no remarkable change after 2:00 p.m. The order of the daily mean $\Psi_L$ value was *C. nepalensis* > *E. decipiens* > *B. papyrifera*.

According to previous experimental results, when the gripping force of the capacitor sensor was 4.1 N, the electrodes were in close contact with the leaf surface without damaging the leaf, and the recorded values remained stable [37]. Therefore, we calculated the corresponding electrophysiological parameters by assuming the gripping force was 4.1 N in this study. The LIWTR values of *C. nepalensis* between 10:00 a.m. and 12:00 p.m. were significantly higher than those at 4:00 and 6:00 p.m., and we found no significant change before 2:00 p.m. (Table 2). We observed no obvious change among the values for *B. papyrifera*. The values for *E. decipiens* at 10:00 a.m. and 2:00 p.m. were significantly higher than at 12:00, 4:00, and 6:00 p.m. The order of the daily mean LIWTR value was *E. decipiens* > *B. papyrifera* > *C. nepalensis*.

The LIWHC values for *C. nepalensis* and *E. decipiens* between 8:00 a.m. and 2:00 p.m. were not significantly different, and the highest values for these two plant species were both at 4:00 p.m. (Table 2). The values for *B. papyrifera* significantly decreased at 10:00 a.m., and then showed no clear change between 10:00 a.m. and 4:00 p.m. The order of the daily mean LIWHC value was *C. nepalensis* > *E. decipiens* > *B. papyrifera*.

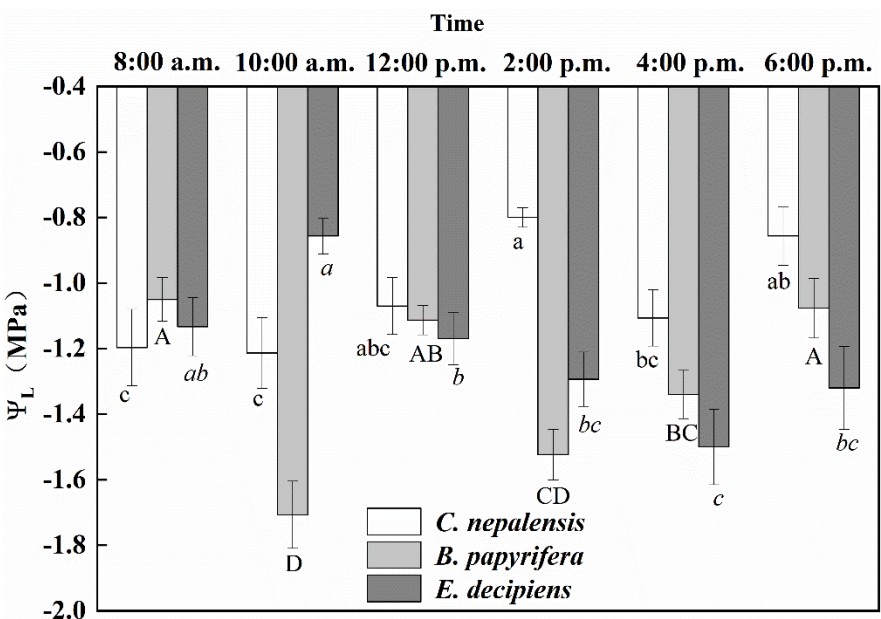

**Figure 1.** Leaf water potential ($\Psi_L$, MPa) of three plant species. Different lowercase letters above the error bars of the same plant species indicate that subsequent values significantly differed at $p \leq 0.05$, according to one-way ANOVA. a, b, and c indicate the differences between the values of *C. nepalensis*; A, B, C, and D indicate the differences between the values of *B. papyrifera*; a, b, and c indicate the differences between the values of *E. decipiens*.

**Table 2.** Leaf intracellular water transport rate (LIWTR, $M\Omega \cdot N^{-1}$), leaf intracellular water-holding capacity (LIWHC), and leaf intracellular water-use efficiency (LIWUE) of *C. nepalensis, B. papyrifera*, and *E. decipiens*. Data are shown as the means $\pm$ SE ($n = 5$). Different lowercase letters in the same column indicate significant differences between different timepoints.

| Plant Species | Time | LIWTR | LIWHC | LIWUE |
|---|---|---|---|---|
| *C. nepalensis* | 8:00 a.m. | 0.24 ± 0.04 [abc] | 288.38 ± 31.05 [b] | 0.05 ± 0.00 [bc] |
| | 10:00 a.m. | 0.41 ± 0.05 [a] | 235.40 ± 31.40 [b] | 0.06 ± 0.01 [ab] |
| | 12:00 p.m. | 0.43 ± 0.17 [a] | 287.49 ± 101.52 [b] | 0.09 ± 0.02 [a] |
| | 2:00 p.m. | 0.35 ± 0.06 [ab] | 226.67 ± 41.76 [b] | 0.07 ± 0.00 [ab] |
| | 4:00 p.m. | 0.04 ± 0.01 [c] | 3039.45 ± 673.30 [a] | 0.02 ± 0.00 [c] |
| | 6:00 p.m. | 0.09 ± 0.03 [bc] | 739.56 ± 175.68 [b] | 0.04 ± 0.01 [bc] |
| *B. papyrifera* | 8:00 a.m. | 0.25 ± 0.03 [a] | 371.46 ± 26.46 [a] | 0.11 ± 0.02 [c] |
| | 10:00 a.m. | 0.29 ± 0.02 [a] | 58.01 ± 9.22 [c] | 0.56 ± 0.12 [a] |
| | 12:00 p.m. | 0.46 ± 0.04 [a] | 126.41 ± 18.04 [bc] | 0.20 ± 0.03 [bc] |
| | 2:00 p.m. | 0.27 ± 0.08 [a] | 65.69 ± 9.44 [c] | 0.34 ± 0.06 [b] |
| | 4:00 p.m. | 0.46 ± 0.15 [a] | 49.91 ± 0.00 [c] | 0.75 ± 0.00 [a] |
| | 6:00 p.m. | 0.31 ± 0.01 [a] | 183.39 ± 50.75 [b] | 0.23 ± 0.09 [bc] |
| *E. decipiens* | 8:00 a.m. | 0.89 ± 0.25 [abc] | 198.25 ± 25.86 [bc] | 0.06 ± 0.01 [ab] |
| | 10:00 a.m. | 1.03 ± 0.20 [ab] | 123.98 ± 39.73 [c] | 0.13 ± 0.06 [a] |
| | 12:00 p.m. | 0.52 ± 0.10 [bcd] | 220.49 ± 24.54 [bc] | 0.06 ± 0.01 [ab] |
| | 2:00 p.m. | 1.18 ± 0.19 [a] | 90.41 ± 15.75 [c] | 0.12 ± 0.02 [ab] |
| | 4:00 p.m. | 0.20 ± 0.06 [d] | 534.29 ± 82.44 [a] | 0.04 ± 0.00 [b] |
| | 6:00 p.m. | 0.47 ± 0.11 [cd] | 341.94 ± 54.26 [b] | 0.05 ± 0.00 [ab] |

The LIWUE values for *C. nepalensis* gradually increased between 8:00 a.m. and 12:00 p.m. and then decreased (Table 2). The values for *B. papyrifera* notably increased at 10:00 a.m., and then decreased at 12:00 and 2:00 p.m.; that at 4:00 p.m. showed no clear difference from that at 10:00 a.m. The LIWUE of *E. decipiens* exhibited no clear change before 2:00 p.m., but the

values at 4:00 and 6:00 p.m. were not remarkably different from that at 2:00 p.m. The order of the daily mean LIWUE value was *B. papyrifera* > *E. decipiens* > *C. nepalensis*.

### 3.2. Diurnal Variations in Photosynthetic Parameters and Instantaneous Water-Use Efficiency

We observed no remarkable change among the $P_N$ values of *C. nepalensis* before 12:00 p.m., but it considerably decreased over time after 2:00 p.m., and the daily mean value was 12.60 μmol·m$^{-2}$·s$^{-1}$ (Figure 2A). The $P_N$ of *B. papyrifera* was much higher between 10:00 a.m. and 2:00 p.m., and then decreased at 6:00 p.m., the daily mean value was 14.99 μmol·m$^{-2}$·s$^{-1}$. The lower $P_N$ of *E. decipiens* was associated with increasing time; the values between 12:00 and 4:00 p.m. were not noticeably different. The daily mean value was 8.34 μmol·m$^{-2}$·s$^{-1}$.

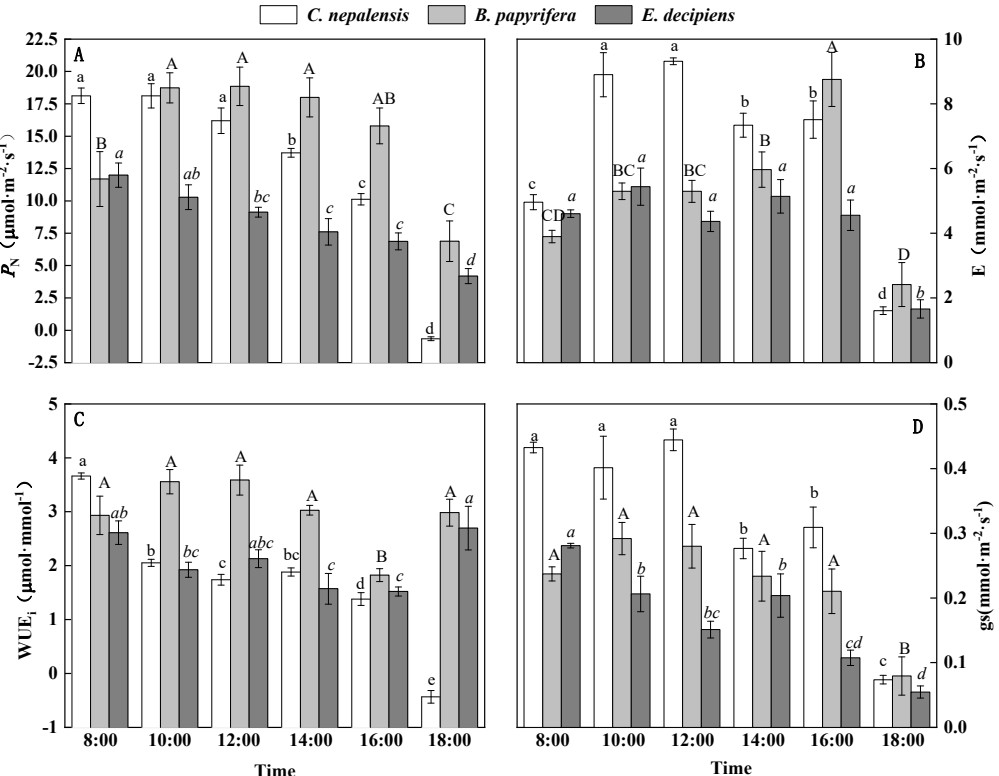

**Figure 2.** Diurnal variations in (**A**) net photosynthetic rate ($P_N$, μmol·m$^{-2}$·s$^{-1}$); (**B**) transpiration rate (E, mmol·m$^{-2}$·s$^{-1}$); (**C**) instantaneous water-use efficiency (WUE$_i$, μmol·mmol$^{-1}$); (**D**) stomatal conductance (gs, mmol·m$^{-2}$·s$^{-1}$). Different lowercase letters appear above error bars of same plant species when subsequent values significantly differed at $p \leq 0.05$ according to one-way ANOVA. a, b, c, etc., indicate differences between values of *C. nepalensis*; A, B, C, etc., indicate the differences between values of *B. papyrifera*; *a, b, c*, etc., indicate the differences between values of *E. decipiens*.

The E values of *C. nepalensis* at 10:00 a.m. and 12:00 p.m. were significantly higher than those at other times; we observed the lowest value at 6:00 p.m. (Figure 2B). The E of *B. papyrifera* did not remarkably change between 10:00 a.m. and 2:00 p.m., and it reached the highest value at 4:00 p.m. The E of *E. decipiens* did not significantly differ between 8:00 a.m. and 4:00 p.m., but decreased to the lowest value at 6:00 p.m.

The WUE$_i$ of *C. nepalensis* decreased over time; the values at 12:00 and 2:00 p.m. were not remarkably different (Figure 2C). The WUE$_i$ of *B. papyrifera* was not notably different between 8:00 a.m. and 2:00 p.m., but was much higher than at 4:00 p.m.; that at 6:00 p.m. was not considerably different from that at 2:00 p.m. The WUE$_i$ of *E. decipiens* gradually decreased from 8:00 a.m. to 4:00 p.m. but did not remarkably change; it noticeably increased at 6:00 p.m.

The gs of *C. nepalensis* between 8:00 a.m. and 12:00 p.m. did not notably differ, and the lowest value appeared at 6:00 p.m. (Figure 2D). We found no significant difference among the gs values of *B. papyrifera* between 8:00 a.m. and 4:00 p.m., and the value at 6:00 p.m. was the lowest. We observed no significant change among the gs values of *E. decipiens* from 10:00 a.m. to 2:00 p.m., and we observed the lowest value at 6:00 p.m.

### 3.3. Diurnal Variations in Chlorophyll Fluorescence Parameters

The $F_o$ value of *C. nepalensis* was the highest at 12:00 p.m., and then exhibited no significant differences between 2:00 and 6:00 p.m. (Figure 3A). We observed higher $F_o$ values in *B. papyrifera* at 8:00 a.m., 2:00, and 6:00 p.m., but these values did not significantly differ. The $F_o$ of *E. decipiens* was the highest at 2:00 p.m. and then notably decreased over time.

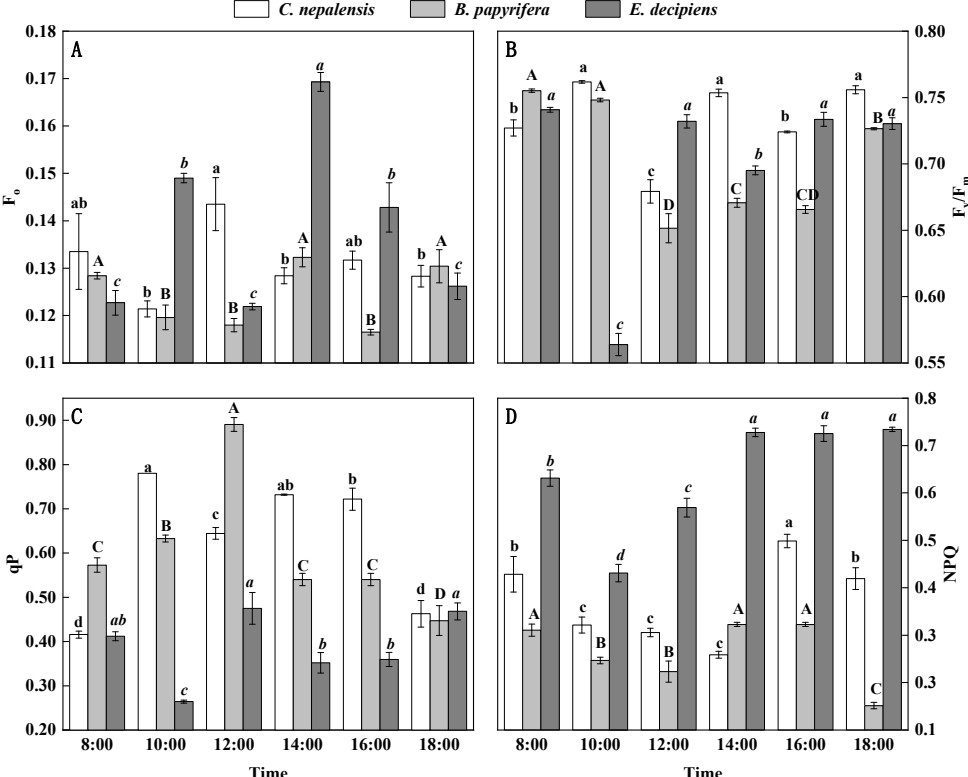

**Figure 3.** (**A**) Minimal fluorescence ($F_o$); (**B**) primary light energy conversion efficiency ($F_v/F_m$); (**C**) photochemical quenching(qP); (**D**) non-photochemical quenching (NPQ). Different lowercase letters appear above error bars of the same plant species when subsequent values significantly differ at $p \leq 0.05$ according to one-way ANOVA. A, b, c, etc., indicate differences between values of *C. nepalensis*; A, B, C, etc., indicate differences between values of *B. papyrifera*; *a, b, c,* etc., indicate differences between values of *E. decipiens*.

We observed higher $F_v/F_m$ values in *C. nepalensis* at 10:00 a.m., 2:00, and 6:00 p.m., and the value at 12:00 p.m. was the lowest (Figure 3B). The $F_v/F_m$ values of *B. papyrifera* between 8:00 and 10:00 a.m. were significantly higher than at other times, and those between 12:00 and 4:00 p.m. were lower than at 6:00 p.m. We observed the lowest $F_v/F_m$ value for *E. decipiens* at 10:00 a.m., and we found no significant difference between the values at 12:00, 4:00 and 6:00 p.m.

The qP values of *C. nepalensis* between 10:00 a.m. and 4:00 p.m. were significantly higher than those at 8:00 a.m. and 6:00 p.m. (Figure 3C). The qP values of *B. papyrifera* increased over time before 12:00 p.m., and then remarkably decreased at 2:00 p.m. The lowest qP value of *E. decipiens* occurred at 10:00 a.m., and the value at 12:00 p.m. was much higher than at 2:00 and 4:00 p.m.

The NPQ values of *C. nepalensis* considerably decreased at 10:00 a.m., and then showed no remarkable change between 10:00 a.m. and 2:00 p.m. The value at 4:00 p.m. was the highest (Figure 3D). The NPQ values of *B. papyrifera* at 10:00 a.m. and 12:00 p.m. were much lower than at 2:00 and 4:00 p.m. We noted the lowest NPQ value of *E. decipiens* at 10:00 a.m. The values between 2:00 and 6:00 p.m. exhibited no clear difference but were higher than those at other times.

*3.4. Difference among Intracellular Water and Photosynthesis in C. nepalensis, B. papyrifera, and E. decipiens*

*C. japonica* exhibited high LIWHC, $P_N$, and E, but low LIWTR, LIWUE, and $WUE_i$ (Table 3). *B. papyrifera* showed a high LIWUE, $P_N$, and $WUE_i$, but low LIWTR, LIWHC, and E. *E. sinensis* showed a relatively high LIWTR, medium LIWHC, and low LIWUE, $P_N$, E, and $WUE_i$.

**Table 3.** Differences among intracellular water use (leaf intracellular water transport rate (LIWTR), water-holding capacity (LIWHC), and water-use efficiency (LIWUE)) and photosynthesis (net photosynthetic rate ($P_N$), transpiration rate (E), and instantaneous water-use efficiency ($WUE_i$)) in *C. nepalensis, B. papyrifera*, and *E. decipiens*.

| Plant Species | Leaf Intracellular Water Traits | | | Photosynthesis | | |
|---|---|---|---|---|---|---|
| | LIWTR | LIWHC | LIWUE | $P_N$ | E | $WUE_i$ |
| *C. nepalensis* | low | high | low | high | high | low |
| *B. papyrifera* | low | Low | high | high | low | high |
| *E. decipiens* | high | middle | low | low | low | low |

## 4. Discussion
### 4.1. Intracellular Water Use vs. Photosynthesis

Plant electrophysiology has been successfully used to study intracellular leaf water [14,23]. Further exploring the photosynthetic response mechanism under adversity is required by jointly analyzing the photosynthetic and electrophysiological traits [24]. In this study, we aimed to investigate the intracellular water transport and use strategies of the three plant species in a karst environment, and to analyze the mechanism through which those strategies influence photosynthetic characteristics. Plants can adapt to karst environments by applying different photosynthetic patterns [38]. *C. nepalensis* is a deciduous plant that have higher root water uptake ability than *E. decipiens*, which is an evergreen plant [39]. The abundant roots of *C. nepalensis* plants also supported their strong water uptake. The high LIWHC in *C. nepalensis* was conducive to maintaining high levels of photosynthesis when the stomata were opened in the morning. Consequently, *C. nepalensis* plants exhibited high photosynthesis, high transpiration, and low $WUE_i$. Unlike *C. nepalensis*, *B. papyrifera* plants efficiently used the inter- and intracellular leaf water, exhibiting high photosynthesis, low transpiration, and high $WUE_i$. This result is consistent with that reported by Li and Wu [40]. Tree species with leathery leaves naturally have a lower water adsorption ability than those with papery leaves [39]. Although the water uptake ability of *E. decipiens* was lower than that of *B. papyrifera*, the leathery leaves helped them store water within leaves and maintain a high LIWTR. *E. decipiens* plants showed a consistently low inter- and intracellular leaf water-use efficiency, and exhibited low photosynthesis, low transpiration, and low $WUE_i$. Therefore, *B. papyrifera* exhibited better adaptability to the karst environment, but *E. decipiens* showed lower water use and photosynthetic efficiency compared with *C. nepalensis* and *B. papyrifera*.

### 4.2. Dynamic Traits of Leaf Intracellular Water

The dynamic traits of leaf intracellular water differed among plant species. From 8:00 to 10:00 a.m., the leaf water loss in *C. nepalensis* enhanced the transport of intracellular water, which was conducive to maintaining the LIWUE and photosynthesis (Figure 4). Therefore,

the water was retained within the cells, which was also attributed to the developed leaf cuticle and strong water uptake ability of the root system, which has many horizontal and oblique roots [39]. *C. nepalensis* plants had a low $\Psi_L$ value, favoring the maintenance of water movement from soil to plant [41]. The higher transpiration and photosynthesis of *B. papyrifera* improved inter- and intracellular leaf water use and consumption. However, the LIWTR of *B. papyrifera* remained stable, which was attributed to the strong water uptake ability of the shallow root system, which had a wide lateral distribution. Consequently, the PSII reaction center maintained stable. The LIWTR, LIWHC, and LIWUE of *E. decipiens* remained stable due to the unchanged transpiration and photosynthesis; the deep root system and leathery leaves also prevented the dissipation of water from the leaf surface [31]. However, the activity of the PSII reaction center of *E. decipiens* was influenced. In this period, *C. nepalensis* showed higher water transport within cells than *B. papyrifera* and *E. decipiens*, whereas *B. papyrifera* efficiently used the intracellular water.

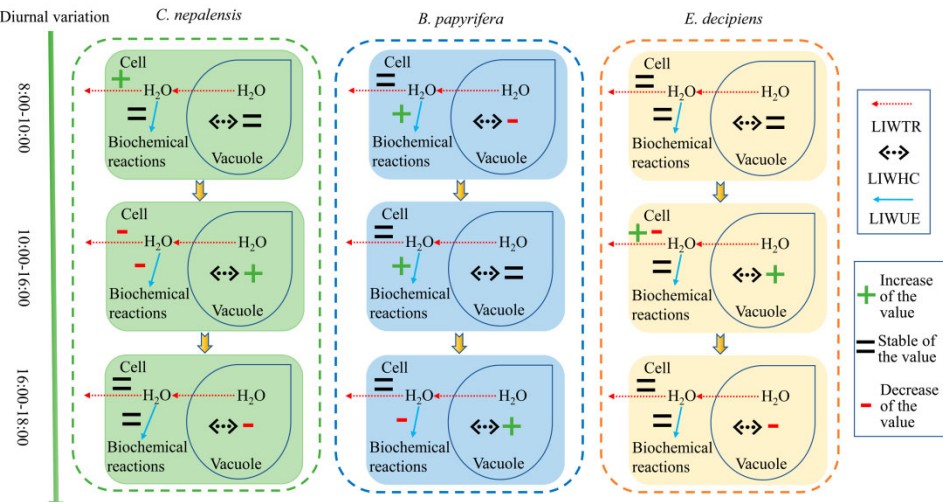

**Figure 4.** Dynamic traits of leaf intracellular water in *C. nepalensis*, *B. papyrifera*, and *E. decipiens*. LIWTR, leaf intracellular water transport rate; LIWHC, leaf intracellular water-holding capacity; LIWUE, leaf intracellular water-use efficiency.

From 10:00 a.m. to 4:00 p.m., water was mainly retained in cells; the transport and use of inter- and intracellular leaf water was limited, which depressed the photosynthesis of *C. nepalensis*. The strong water-holding capacity of cells helped to maintain high light energy use efficiency in *C. nepalensis*, which was indicated by the variations in qP and NPQ [42]. The strong water uptake ability of the roots and notably decreased water consumption of *C. nepalensis* increased the intracellular water storage at 4:00 p.m., which also helped *B. papyrifera* to maintain a stable inter- and intracellular water status. The increased use of intracellular water supported its high photosynthetic efficiency. The remarkably decreased $F_v/F_m$ in *B. papyrifera* indicated that the photosynthetic structure was damaged, and the plants suffered from water deficit [22]. However, the conversion and use of light energy were increased to maintain the stability of photosynthesis [20]. We mainly attributed the fluctuation in the LIWTR of *E. decipiens* in this period to the slightly changed transpiration, whereas the slowly declined photosynthesis reduced the water consumption within the leaves and increased the LIWHC. The stability of the photosynthetic structure and the activity of PSII reaction center recovered to the level at 8:00 a.m. due to the conservation of intracellular water [43]. In this period, we mainly observed increased water-holding capacities for the leaves of *C. nepalensis* and *E. decipiens*, but *B. papyrifera* exhibited higher intracellular water-use efficiency and photosynthetic capacity than *C. nepalensis* and *E. decipiens*.

From 4:00 to 6:00 p.m., the closed stomatal aperture decreased the transpiration pull and kept the water within cells [15]. The reduction in the $P_N$ of *C. nepalensis* mainly occurred

owing to stomatal closure in the afternoon, which inhibited the transpiration and retained the water within the leaves [44]. As a result, the activity of PSII reaction center recovered. Although lower at 6:00 p.m., the $P_N$ of *B. papyrifera* was still not lower than those of *C. nepalensis* and *E. decipiens* at 4:00 p.m. The water consumption and $WUE_i$ of *B. papyrifera* remained high, whereas the decreased use of intracellular water increased the LIWHC of *B. papyrifera*. Consequently, the stability of the PSII reaction center in *B. papyrifera* also recovered. The considerably increased $WUE_i$ and stable LIWUE slowed the decrease in the photosynthesis in *E. decipiens*, whereas the lower water uptake ability by roots compared with that of the other two plant species decreased the LIWHC. The water status in the three plant species all entered to the recovery phase after 6:00 p.m. Water transport within the leaf cells of *C. nepalensis*, *B. papyrifera*, and *E. decipiens* stabilized in this period.

Consequently, *B. papyrifera* will be more adaptable to the karst drought caused by future climate change than the other two plant species, because it can alternatively and efficiently use the inter- and intracellular leaf water with changing surroundings. *C. nepalensis* can efficiently conserve the intracellular leaf water and exhibits an efficient photosynthetic capacity, which helps this species to better adapt to the karst droughts than *E. decipiens*.

## 5. Conclusions

Plants adjust their photosynthesis by changing the water transport and use within leaf cells in karst environments. With strong water uptake ability by the roots and high leaf intracellular water-holding capacity, *C. nepalensis* plants maintained high photosynthesis, and the stomata apertures quickly closed to conserve intracellular water. *C. nepalensis* plants exhibited high photosynthesis, high transpiration, and low $WUE_i$. *B. papyrifera* maintained the stable transport of intracellular water, and their increased intracellular water-use efficiency improved their photosynthetic efficiency. The photosynthesis of *B. papyrifera* was also maintained by efficiently using the transpired water when the intracellular water-holding capacity increased. *B. papyrifera* showed high photosynthesis, low transpiration, and high $WUE_i$. The inter- and intracellular leaf water of *E. decipiens* remained stable due to its strong water-holding capacity, which could be attributed to the leathery leaves. The photosynthesis of *E. decipiens* remained stable, but the photosynthetic structure and PSII reaction center were notably influenced. *E. decipiens* exhibited low photosynthesis, low transpiration, and low $WUE_i$. *B. papyrifera* exhibited better adaptability to the karst drought than *C. nepalensis* and *E. decipiens*. Our results provide a reference for accurately analyzing plant photosynthetic adaptability in heterogeneous karst environments, and they can be used for improving the revegetation efficiency in these fragile habitats.

**Supplementary Materials:** The following supporting information can be downloaded at https://www.mdpi.com/article/10.3390/agronomy12112758/s1. Supplementary File 1: The calculation of leaf intracellular water transport rate, water-holding capacity and water-use efficiency based on electrophysiological parameters.

**Author Contributions:** Conceptualization, D.X. and Y.W.; methodology, Y.W., D.X., and X.Q.; data curation, X.Q. and W.W.; writing—original draft preparation, X.Q. and D.X.; writing—review and editing, M.L. and K.S.; funding acquisition Y.W. All authors read and agreed to the published version of the manuscript.

**Funding:** This research was funded by the Support Plan Projects of Science and Technology of Guizhou Province [No. (2021) YB453], the National Key Research and Development Program of China [No. 2021YFD1100300], and the Priority Academic Program Development (PAPD) of Jiangsu Higher Education Institutions.

**Data Availability Statement:** The datasets analyzed during the current study are available from the corresponding author upon reasonable request.

**Conflicts of Interest:** The authors declare no conflict of interest.

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
