# Peer review of "Diurnal Variation in Transport and Use of Intracellular Leaf Water and Related Photosynthesis in Three Karst Plants"

_agronomy, doi:10.3390/agronomy12112758_

Round 1
Reviewer 1 Report (Previous Reviewer 1)
Authors incorporated all corrections/ suggestions nicely. The MS can be acceptable for publication.
Author Response
Dear Reviewer,
Thank you very much for your positive and constructive comments and suggestions on our manuscript.
We have revised the Results and Discussion parts to make them more clearly stated. Furthermore, we have conducted an English language editing from the services provided by the MDPI. The revisions using "track changes" can be seen in the revised manuscript.
All the best
Deke
Reviewer 2 Report (Previous Reviewer 2)
Since the main methodological problems previously highlighted were mainly due to a poor writing, the manuscript has now acquired significance. Authors have correctly measured both gas exchange and water potential parameters. Moreover, they used a sufficient number of biological replicates. I still believe that manuscript is poorly written in some parts, especially results and discussione, and difficult to be read. Even if the aim of the work is clearly presented in the introduction, the discussion is still fragmented and the differences between plant species are little integrated, I suggest to try to simplify the discussione avoiding to repeat and detail the results.
I report some minor suggestions in the attached file.
I would like to thank authors for their explanation about the water potential instrument functioning.

Author Response
Please see the attachment.

This manuscript is a resubmission of an earlier submission. The following is a list of the peer review reports and author responses from that submission.
Round 1
Reviewer 1 Report
The authors assessed the impact of diurnal changes of transport and utilization of leaf intracellular water and the related photosynthesis responses in three karst plants. Overall, the present article is well drafted, statistically sound, and the results are fine in relation to photosynthetic leaf gas exchange vs. diurnal dynamics.
Few minor comments/ suggestions are as stated below:
· L 13-15: What is the plant growth age of selected plants?
· L 122: Plz mention the author year (Hang et al. (??)
· L 123-124: Please mention experiment (data) observation month and year for clarity
· L 124: High temperature??
· L 156-161: Please incorporate photosynthetic photon flux density (PPFD), air temperature, and CO2 level??
· For tables: Plz correct the presentation of the different letters means superscript values
· Figures 2 and 3: Please remove Figures A and B X-axis scale. Figure C and D X-axis scale is sufficient to present
· L 403: Please remove the repeated reference serial numbers.
· English should be improved throughout the MS.
Reviewer 2 Report
Qin et al. have measured some leaf eco-physiological parameters to assess the adaptability of three species widely distributed in karst region to water stress. I believe that beyond underline the differences among plant species, a real discussion about which species is predicted to be more tolerant to the future climate change scenarios is lacking. I suggest to further improve this part adding some speculations about which could be the more resistant species in relation to the drought predicted climate in this region.
Beyond this general comment, the manuscript presents some serious methodological issues. First, authors used leaves detached from branches and rehydrated for 30 minutes. This processing surely does not allow to obtain real measures of gas exchanges or water potential, especially to asses diurnal variations. An other concerns is about the biological replicates, it is not clear if authors used 5 plants of each species or 5 leaves from one single plant for each species. This aspect is crucial since if only one plant has been used the reliability of results is too low making the manuscript unacceptable to drive any conclusion. Also how they perform three measures for each parameter on each leaf must be detailed (it is hard to not damage a leaf with a so high number of measures). I also would like to know if the instrument used for measuring the water potential has been validated with a pressure chamber.
I reported other comments in the attached file.
